# Exploring extreme environments in Türkiye for novel P450s through metagenomic analysis

Hande Mumcu[1,2☯], Julian Zaugg[3☯], Irem Keles[1,2], Aycan Kayrav[1,2], Nurgul Balci[4], David R. Nelson[5], Philip Hugenholtz[3], Elizabeth M. J. Gillam[6], Nevin Gul Karaguler[1,2*]

1 Department of Molecular Biology and Genetics, Faculty of Science and Letters, Istanbul Technical University, Istanbul, Türkiye, 2 Dr. Orhan Öcalgiray Molecular Biology-Biotechnology and Genetics Research Center, Istanbul Technical University, Istanbul, Türkiye, 3 Australian Centre for Ecogenomics, School of Chemistry and Molecular Biosciences, The University of Queensland, Brisbane, Queensland, Australia, 4 Geomicrobiology-Biogeochemistry Laboratory, Department of Geological Engineering, Istanbul Technical University, Istanbul, Türkiye, 5 The University of Tennessee Health Science Center, Memphis, Tennessee, United States of America, 6 School of Chemistry and Molecular Biosciences, The University of Queensland, Brisbane, Queensland, Australia

☯ These authors contributed equally to this work.
* karaguler@itu.edu.tr

## Abstract

Cytochrome P450 enzymes (P450s), particularly those of microbial origin, are highly versatile biocatalysts capable of catalyzing a broad range of regio- and stere-oselective reactions. P450s derived from extremophiles are of particular interest due to their potential tolerance to high temperature, salinity, and acidity. This study aimed to identify and classify novel microbial P450 enzymes from extreme environments across Türkiye, including hydrothermal springs, hypersaline lakes, and an acid-mine drainage site. The focus of this study was on classifying the sequence diversity of P450 enzymes in these sites. To that end, shotgun metagenomic analysis of six sites, using de novo binning, phylogenetic analysis, and functional gene anno-tation, was used to discover 311 putative P450 sequences, assigned to 87 families and 158 subfamilies, including 8 novel families and 49 new subfamilies. Of these, 237 were in 138 metagenomic bins, including 45 high-quality metagenome-assembled genomes. The distribution of P450 families varied across sites, reflecting distinct environmental conditions and microbial community compositions. These findings highlight the untapped potential of Türkiye's extreme habitats as a source of novel biocatalysts. Beyond their industrial relevance, extremophile-derived P450s may also play key roles in enabling microbial adaptation to harsh environmental conditions, through their involvement in stress-responsive metabolic pathways and structurally resilient enzyme forms. This work provides a foundation for future studies into both their biotechnological applications and ecological functions.

**Data availability statement:** The shotgun sequencing read data for the samples described in this study, as well as the 171 higher quality MAGs obtained from them, have been deposited in the NCBI Sequence Read Archive (SRA) (Accessions: SRR27869035–SRR27869040) and Genbank, respectively, under the bioproject accession PRJNA979897 (https://www.ncbi.nlm.nih.gov/bioproject/PRJNA979897/).

**Funding:** 1- NGK, Grant No. 1059B192100859, The Scientific and Technological Research Council of Turkey (TUBITAK), https://tubitak.gov.tr/en 2- NGK, Grant Nos. 42997 and 42953, ITU Scientific Research Projects Division, https://bap.itu.edu.tr/ The funders had no role in study design, data collection and analysis, decision to publish, or preparation of the manuscript.

**Competing interests:** The authors have declared that no competing interests exist.

## Introduction

Metagenomics is a culture-independent approach for studying microbial communities by extracting and sequencing genetic material directly from environmental samples (eDNA). Unlike traditional microbiology, which relies on cultivating microbes in the laboratory, metagenomics provides a much less biased view of microbial communities, including previously unculturable species [1]. This approach offers unprecedented insights into microbial diversity, metabolic functions, and ecological interactions, enabling researchers to study microorganisms in their natural habitats without the need for isolation [2].

Among metagenomic techniques, shotgun metagenomics has emerged as a powerful tool for exploring the functional capacity of microbial communities. By randomly sequencing genetic material within a sample, this approach enables the identification of novel genes, biosynthetic gene clusters, and entire metabolic pathways [3]. Through the use of the numerous computational tools that have been developed to process such data, it is now possible to reconstruct the genomes of novel microorganisms and functionally annotate their genes, providing researchers with insight into their ecological roles [4]. This approach is instrumental in identifying new enzymes and biomolecules with potential biotechnological applications, including cytochrome P450 enzymes, which play a crucial role in oxidative metabolism across various biological systems [5].

Cytochrome P450 heme-thiolate proteins (EC 1.14.14.1) are a superfamily of enzymes usually acting as monooxygenases. The majority of these enzymes catalyze the insertion of one oxygen atom from molecular oxygen into the substrate, with reduction of the other atom to water, a process facilitated by the presence of one or more redox partners that catalyze electron transfer from the reducing cofactor, NADPH. P450s bind molecular oxygen through their heme prosthetic group that is coordinated to the apoprotein through a conserved axial cysteine residue [6]. Although P450s catalyze different types of reactions, they have a common catalytic cycle consisting of nine steps [6] that involves the transfer of two electrons. The electrons are usually transferred to the heme center through redox protein partners such as ferredoxins/ferredoxin reductases or diflavin reductases in a multi-component electron transfer chain. However, some P450s are present as genetic fusions with one or more redox partners and are therefore considered self-sufficient [7].

To date, many bacterial and archaeal cytochromes P450 have been identified and classified [8–12]. Characterized P450s play roles in many catabolic and anabolic pathways such as fatty acid, steroid, and xenobiotic degradation, and the biosynthesis of primary and secondary metabolites [13,14]. Within those pathways, they act on diverse simple and complex molecules such as fatty acids, alkanes, terpenes, eicosanoids, vitamins, steroids, antibiotics, and a variety of drugs and other xenobiotics [15]. In addition to their wide substrate and reaction diversity, the most important feature of microbial P450s is that they can be regio- and stereo-specific [16]. Consequently, they are useful in synthesizing new drugs, fine and bulk chemicals, and agrochemicals in the pharmaceutical, flavour/fragrance, and agricultural sectors, as

well as for pollutant removal [17]. The extensive intrinsic sequence diversity in microbial P450s and their potential to be used in many industrial processes make them attractive biocatalysts, and the identification of novel P450s is an area of intense interest [18].

Extreme environments, including hydrothermal vents, polar deserts, hypersaline lakes, acidic mines, and deep-sea sediments, host diverse microbial communities, collectively known as extremophiles. These microbes have evolved unique adaptive strategies to survive the harsh conditions characteristic of such environments, e.g., high temperatures, salinity, pH, and concentrations of heavy metals. Extremozymes—enzymes found in extremophiles—enable survival under these conditions and exhibit remarkable stability and activity, making them highly valuable for biotechnological applications [19]. P450 extremozymes, in particular, have garnered significant attention due to their diverse catalytic capabilities, but relatively few have been identified to date. Extremophilic P450s characterized to date include members of the self-sufficient CYP116 family [20], as well as the CYP119, CYP154, CYP174, CYP175, and CYP231 families [21]. Jiang et al. identified three moderately halophilic P450 fatty acid decarboxylases—CYP152L1_ortholog, CYP152L7, and CYP152L8—belonging to the CYP152 family [22]. Moreover, Nguyen and colleagues identified 36 potentially thermostable P450s from water samples collected at Binh Chau hot spring in Vung Tau, Vietnam, through metagenome shotgun sequencing [23]. They also discovered a novel moderately alkali-thermophilic P450 from the CYP203 subfamily, which exhibits optimal activity at 50 °C and pH 8.0 [24].

The climatic conditions at various locations across the Anatolian geography allow different species of living organisms to occupy unique habitats and ecological niches. Türkiye, one of the richest countries in Europe in terms of biodiversity, is home to many endemic species not commonly found elsewhere. The aim of the present study was to characterize the prokaryotic community and P450 diversity of six previously uncharacterized sites in Türkiye with extreme environmental conditions through *de novo* binning, phylogenetic analysis, and functional gene annotation of metagenomic data. This study identified and classified a total of 311 microbial cytochromes P450 across 87 families and 158 subfamilies, including 8 new families and 49 new subfamilies. The findings underscore the value of investigating extreme environments as a rich source of novel and functionally diverse enzymes.

## Materials and methods

### Sampling

Samples were collected from six sites in Türkiye characterized by extreme environmental conditions, with three samples collected from each site (Fig 1) (USGS National Map Viewer): Lake Acıgöl (37.8299 N, 29.8931 E; April 2024 spring) [25], Gömeç (Balıkesir; 39.386373 N, 26.835452 E; July 2019 summer), Hisaralan (Balıkesir; 39.287251 N, 28.341724 E; December 2021 winter), Armutlu (Yalova; 40.520437 N, 28.815628 E; July 2017 summer) [26], Balya (Balıkesir) acid mine drainage (39.749294 N, 27.578101 E; August 2010 summer) [27] and Tuz Gölü (38.818571 N, 33.347851 E; March 2022 spring). Lake Acıgöl, Tuz Gölü, and Gömeç are hyper-saline environments [25,28]. Located in hydrothermal regions, Hisaralan and Armutlu have average water temperatures of 98 °C and 74 °C [26], respectively. Balya acid mine drainage has a pH lower than four and contains high concentrations of sulfur and heavy metals such as Pb, Zn, and Cu [27].

Sediment samples were collected from Lake Acıgöl (the upper 10 cm of the lake bed sediments), Gömeç (the upper 10 cm of the lake bed sediments), Armutlu (at a depth of 10–20 cm of the pool) and Balya (at a depth of 10 cm of the acidic pools); a two-liter water sample was collected from Hisaralan (at a depth of 10–20 cm of the pool); and an approximately 110 g sample of salt crystals was collected from Tuz Gölü. The salt crystals precipitated from the water columns (< 20 cm) were collected from the lakebed. All collections were done in accordance with permits obtained from the Republic of Türkiye Ministry of Environment Urbanization and Climate Change explicitly for the field studies described here.

### Environmental DNA extraction and shotgun metagenomic sequencing

Environmental DNA (e-DNA) was isolated from 0.5–1 g of each sediment sample using the Qiagen DNeasy PowerSoil Pro Kit. The hot spring and saltwater samples were dissolved slowly in 2 L phosphate buffer saline (PBS), filtered through

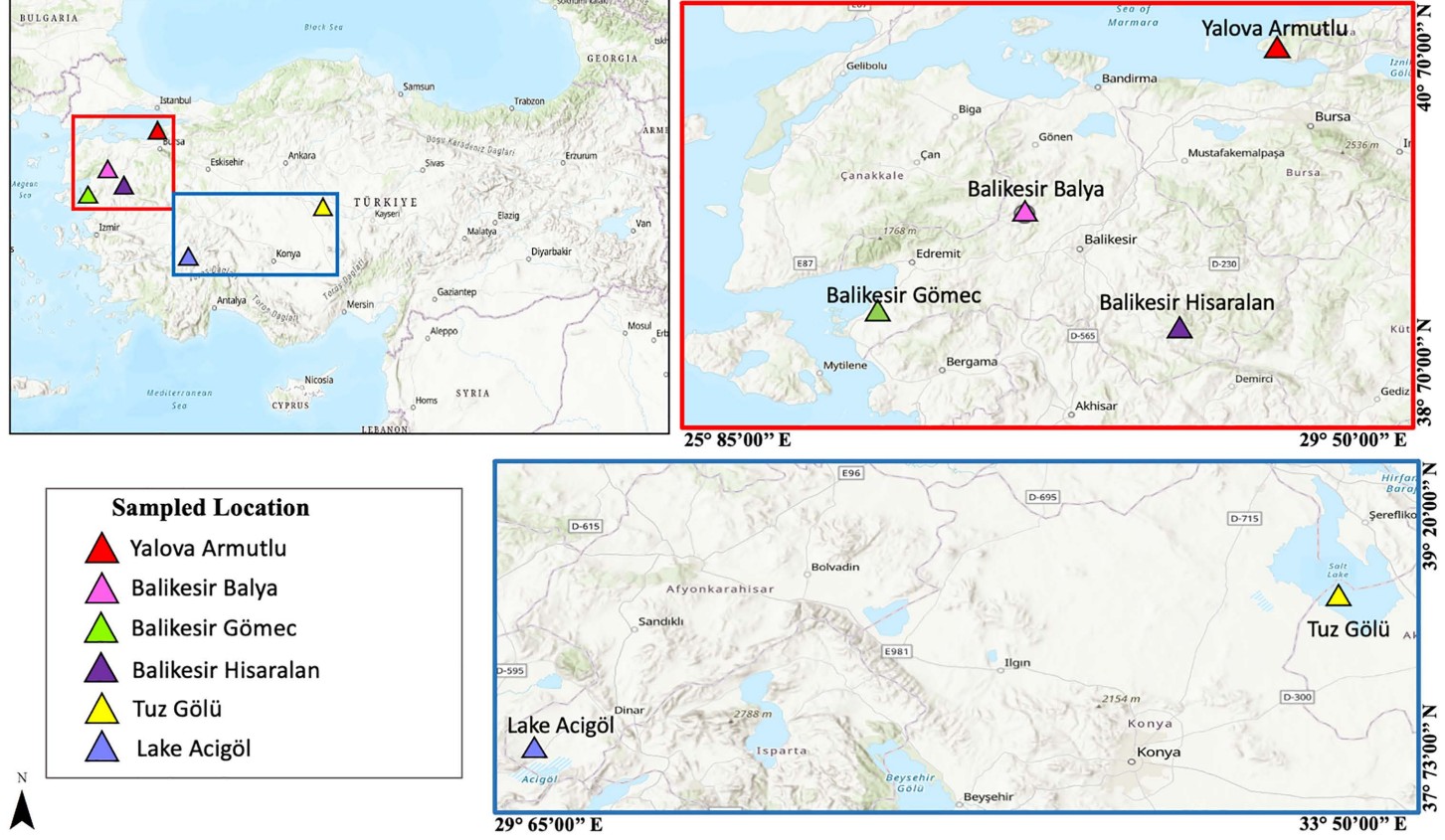

**Fig 1. Maps showing the locations of the six sampling sites in Türkiye (USGS National Map Viewer).**

a 0.22 µm sterile syringe filter with the help of vacuum, and then the e-DNA was isolated using a Qiagen DNeasy Power-Water Kit. DNA purity and quality were assessed using Qubit 2.0 DNA HS Assay (Life Technologies). Shotgun sequencing libraries were prepared using KAPA HyperPrep Kit (Roche) and library concentration and quality control were evaluated using Qubit 2.0 DNA HS Assay (Life Technologies) and Tapestation High Sensitivity D1000 Assay (Agilent Technologies). The 150 bp paired-end sequencing of prepared libraries was performed on an Illumina NextSeq 550 system. An overview of the experimental and computational (see below) methods used to process the samples is provided in Fig 2.

## Metagenomic assembly and de-novo binning

Low quality reads were identified and removed with Trimmomatic (ver. 0.39, ILLUMINACLIP: NexteraPE-PE:2:30:10, SLIDINGWINDOW:4:15, MINLEN:50) [29]. Quality controlled reads were then assembled using metaSPAdes (ver. 3.15.4) [30] with default parameters. Quality controlled reads for each sample were mapped onto their respective scaffolds with minimap2 (ver. 2.17) [31] using the 'make' mode in the DNA read coverage calculator CoverM (ver. 0.6.1) [32]. Low quality read mappings were removed with the CoverM 'filter' mode (minimum identity 95% and minimum aligned length of 75%), and the number of remaining reads was used to calculate the fraction of the DNA mapping to the assembled scaffolds.

The assembly for each sample was binned using the metagenomic binning pipeline Aviary (ver. 0.5.6) [33]. Briefly, Aviary first maps reads from all samples to each individual assembly with minimap2 (ver. 2.17) as part of CoverM (ver. 0.6.1) to obtain differential coverage information for each assembly. Using this coverage information, metagenome contigs

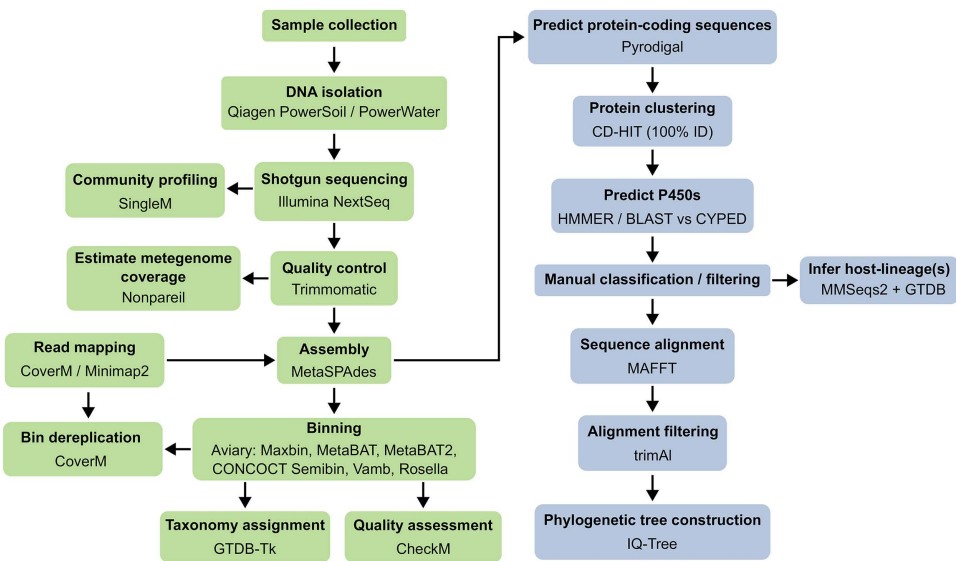

**Fig 2. Schematic showing the processing steps performed in the present study.**

were then binned using the Maxbin (ver. 2.2.7) [34], MetaBAT (ver. 0.32.5) [35], MetaBAT2 (ver. 2.15) [36], CONCOCT (ver. 1.1.0) [37], Vamb (ver. 3.0.2) [38], Semibin (ver. 1.1.1) [39] and Rosella (ver. 0.4.2) [40] binning methods with a minimum contig length of 1,500 bp and minimum bin size of 200,000 bp. For each sample, an optimal, non-redundant set of bins produced from the various binning tools were selected by DAS Tool (ver. 1.1.2) [41]. The completeness and contamination of all 1,138 non-redundant bins were calculated by CheckM (ver. 1.1.3) [42]. Taxonomy was assigned to each bin using the Genome Taxonomy Database Toolkit (GTDB-Tk; ver. 2.3.0; with reference to GTDB R08-RS214) [43,44]. The non-redundant bins from across all samples were then clustered and dereplicated using CoverM 'cluster' (precluster-method = dashing) with an ANI threshold of 97% and accounting for bin quality (checkm-tab-table). Dereplication yielded 1,135 bins, 171 of which were higher quality with a quality value ≥ 50 (calculated as the completeness – (3 × contamination).

## Metagenome community profiling

The relative abundance of the dereplicated bins was calculated by first mapping the reads from each sample to each using CoverM 'make' and removing low quality mappings with CoverM 'filter' (minimum identity 95% and minimum aligned percent of 75%). The mean coverage of each bin was then calculated with CoverM and the relative abundance of each, among those obtained, was calculated as its coverage divided by the total summed coverage of all bins (S1 Table).

To obtain a broader assessment of the community composition of each sample, the microbial community profiler SingleM (ver. 0.16.0) was used [45]. Taxonomic profiling tools typically rely on databases derived from reference genomes [46–50], limiting abundance calculations to known species while missing novel taxa [45]. In contrast, SingleM can identify lineages where no genome exists. Briefly, it achieves this by a) analyzing only those reads which cover highly conserved regions of single copy marker genes, b) clustering these reads *de novo* into operational taxonomic units (OTUs), independent of existing taxonomies, c) taxonomically classifying OTUs against the Genome Taxonomy Database (GTDB) [51,52], d) per marker gene, estimating the relative abundance of each taxon based on OTU classifications, and e) calculating a trimmed mean abundance taken across all the marker genes [45]. The bacterial and archaeal community composition of each sample was therefore determined by classifying those raw reads corresponding to 59 single-copy

genes using the 'pipe' tool from SingleM, based on taxonomies derived from the GTDB R08-RS214. SingleM 'condense' was used to produce a single OTU table containing the trimmed mean coverage across each lineage, calculated across all genes. The relative abundance of each lineage was then calculated as its respective coverage divided by the total summed coverage for each sample. Shannon diversity was calculated for each sample from genus level mean coverage values from SingleM using phyloseq (ver. 1.50.0) [53]. Finally, Nonpareil was run on the quality-controlled reads using the k-mer alignment method to assess the fraction of the microbial community sampled by sequencing [54,55]. Community abundance stacked bar charts were created using the R package ggplot (ver. 3.4.4) [56], and heatmaps with Complex Heatmap (ver. 2.16.0) [57].

### Gene extraction, and identification and classification of P450s

Protein-coding sequences (CDS) in the assembled scaffolds and bins were first predicted using Pyrodigal (ver. 2.0.2) [58], a Python library binding to Prodigal [59], in metagenomic mode. Sequences with start and stop codons, i.e., theoretically complete open reading frames, were extracted using mfqe [60] (ver. 0.5.0). Complete protein sequences (1,966,993) were clustered at 100% protein identity using CD-HIT (ver. 4.8.1) [61], with all members of each cluster required to have at least 80% of their sequence overlapping with the longest (seed) sequence. Protein sequences containing the cytochrome P450 domain (PF00067) were identified using HMMER hmmscan (ver. 3.3.2; -E 1e-5) [62] and by aligning the protein sequences against the CYPED database [63] with DIAMOND blastp (--evalue 0.00001, --query-cover 50, --subject-cover 50, --id 15) [64]. Of the 4,064 putative P450 sequences identified (2,730 BLAST, 1,334 HMMER), 311 were identified as complete P450s after manual inspection. The selected sequences were aligned using MAFFT (--localpair, ver. 7.455) [65], and the resulting alignment trimmed using trimAl (-automated1, ver. 1.4.1) [66]. A phylogenetic tree was then constructed using IQ-Tree (model LG+R7, ver. 2.1.2) [67] with 1,000 bootstraps and visualized using tvBOT [68]. Approximately 117 of the identified P450 sequences were either not found in a genome bin or were found in a bin with a poorly resolved taxonomic classification, i.e., the bin could not be taxonomically classified below the class level. For these sequences, similar sequences were searched for among the representative genomes from the GTDB (R08-R214) using MMseqs (ver. 13.45111; --min-seq-id 0.7 -c 0.7) [69], and the best hit was used to annotate the corresponding host-lineage in the phylogenetic tree.

Proteins within the P450 superfamily are classified in accordance with the guidelines set by the International P450 Nomenclature Committee [6,70]. Specifically, proteins sharing more than 40% sequence similarity were placed within the same family, while those with over 55% sequence similarity were categorized within the same subfamily [71]. Any proteins having less than 40% sequence similarity to known P450s were assigned to a novel P450 family.

### Code availability

This section confirms that all analyses were performed using published and/or publicly available tools.

## Results

### Taxonomic profiling of the extreme sites

Shotgun sequencing produced 18–24 Gbp of read data for each sample, except for the Armutlu hot spring, where 2.4 Gbp was obtained. Estimated coverage of the microbial communities ranged from 35–96% (66–96% excluding Armutlu; S1 Fig and S2 Table), suggesting that a substantial portion of the community was sampled. Dominant phyla (>10% relative abundance in at least one sample) included archaeal lineages from *Halobacteriota* (Tuz Gölü, Gömeç), *Nanohaloarchaeota* (Tuz Gölü) and *Thermoproteota* (Armutlu)*, and bacterial lineages *Actinomycetota* (Hisaralan), *Aquificota* (Hisaralan), *Bacillota* (Hisaralan), *Bacteroidota* (Lake Acıgöl), *Bipolaricaulota* (Hisaralan), *Chloroflexota* (Armutlu) and *Pseudomonadota* (Lake Acıgöl, Armutlu, Balya, Gömeç) (Fig 3, S1–S3 Tables). Notably, *Halobacteriota* are extremely halophilic archaea

[72], *Nanohaloarchaeota* are exclusively derived from hypersaline habitats [73], and *Thermoproteota* are methanogenic and hyperthermophilic archaea (Fig 3). Of the bacteria, members of the *Bipolaricaulota* (15.7%) are known to fix carbon and dominate in some geothermal regions [74]; the family *Thiomicrospiraceae* (15.1% in Balya), belonging to the phylum *Pseudomonadota*, has an important role in sulfur oxidation pathways [75]; and *Bacteroidota* (26.4% in Acigol) is essential for the nitrogen cycle in hypersaline environments and significantly contributes to the elimination of greenhouse gases [76]. The taxonomic composition of these extreme environments reveals a diverse range of archaeal and bacterial lineages, with site-specific differences that may be shaped by distinct selective pressures.

## Metagenome assembled genomes (MAGs)

A total of 1,138 metagenomic bins were obtained, 171 of which were deemed high quality (>50 combined completeness/contamination metric). These 171 MAGs were estimated to represent between 0.6–65% of the microbial communities from which they were derived (S4 Table). They belong to four archaeal phyla: (*Halobacteriota*, n = 21; *Nanoarchaeota*, n = 1; *Nanohaloarchaeota*, n = 7; and *Thermoproteota*, n = 5), and 28 bacterial phyla (*Acidobacteriota*, n = 4; *Actinomycetota*, n = 4; *Aquificota*, n = 2; *Armatimonadota*, n = 2; *Bacillota*, n = 10; *Bacillota*_A, n = 3; *Bacillota*_C, n = 1; *Bacillota*_F, n = 3; *Bacteroidota*, n = 32; *Bipolaricaulota*, n = 1; *Campylobacterota*, n = 1; *Chloroflexota*, n = 9; *CSP1–3*, n = 1; *Cyanobacteriota*, n = 3; *Deinococcota*, n = 3; *Desulfobacterota*, n = 6; *Desulfobacterota*_D, n = 1; *Desulfobacterota*_F, n = 1; DRYD01, n = 2; *Fibrobacterota*, n = 1; *Gemmatimonadota*, n = 1; *Marinisomatota*, n = 1; *Nitrospirota*, n = 1; *Patescibacteria*, n = 5; *Planctomycetota*, n = 3; *Pseudomonadota*, n = 32; *Spirochaetota*, n = 2; *Thermotogota*, n = 1; and, notably, one novel phylum) (Fig 4, S5 and S6 Tables). At lower taxonomic levels, many of the MAGs appear to represent novel lineages: 118 were unclassified at the species level, 22 at the genus, 4 at the family and 2 at or above the order level. A summary of all 1,138 bins is provided in the supplementary material (S5 Table). The recovered MAGs expand our understanding of the genomic diversity in these extreme environments, revealing several novel lineages that warrant further characterization.

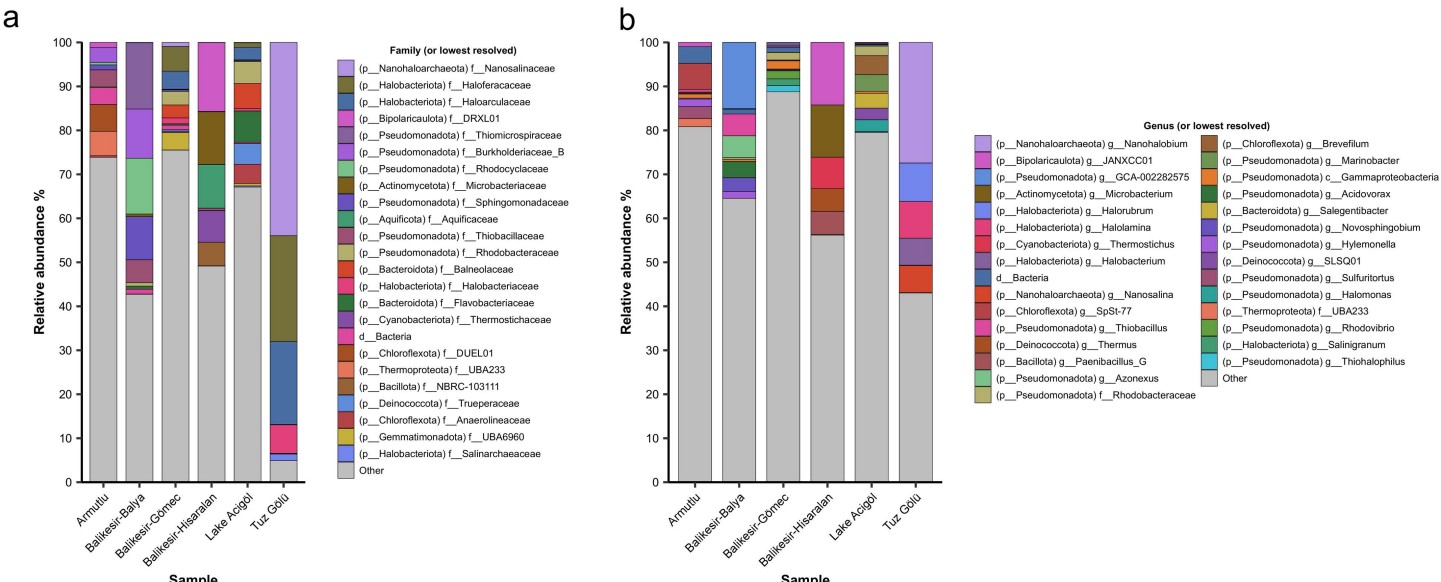

**Fig 3. Stacked bar charts of the prokaryote relative abundance profiles of the six sites at the (a) family and (b) genus levels (or lowest resolved taxonomy level), based on the mean coverage of each lineage as reported by SingleM.** Only the top five families/genera per sample are shown, with all other taxa grouped under 'Other'.

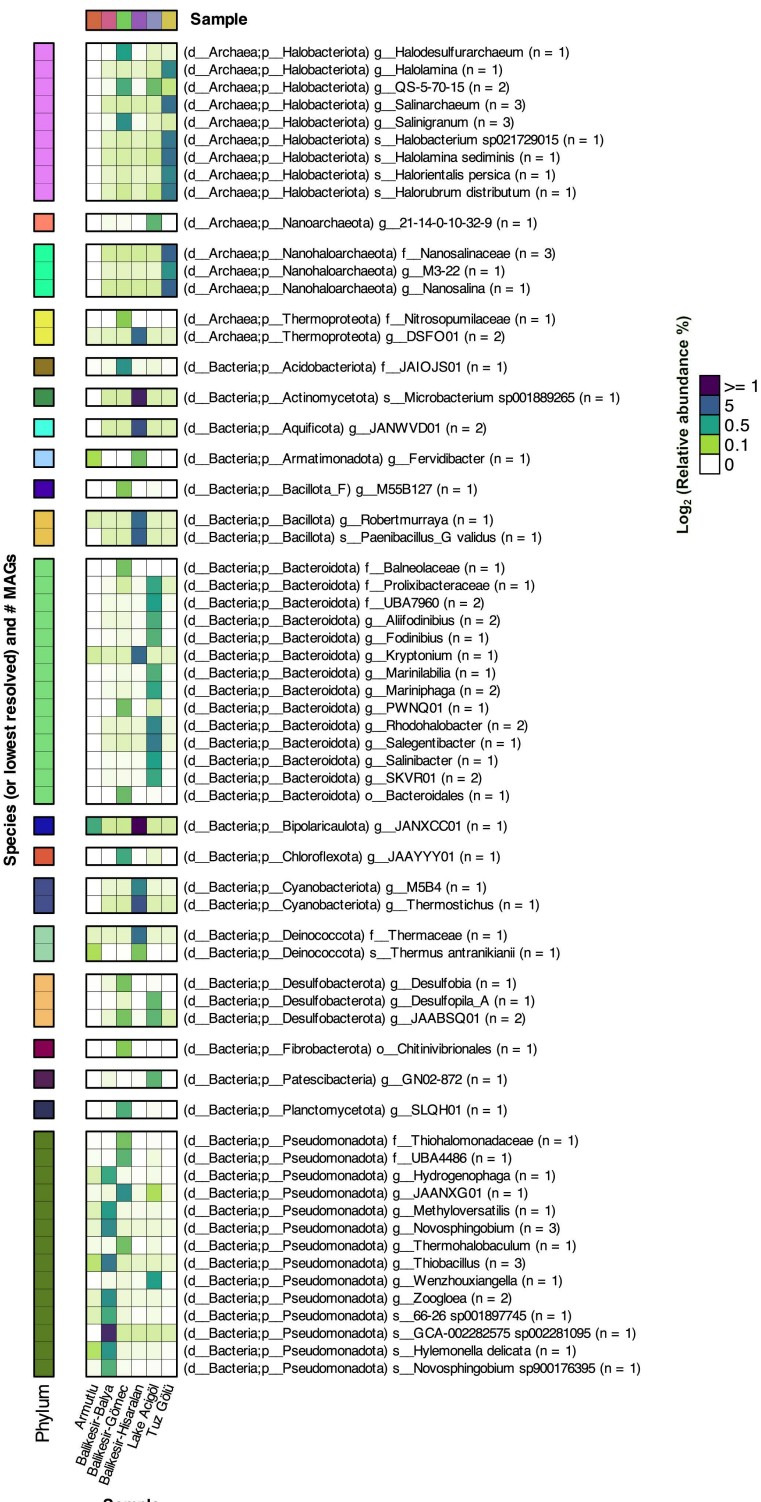

**Fig 4. Heatmap showing the relative abundances of species (or lowest resolved taxonomic rank) with an abundance of at least 2% in at least one of the six samples, based on the abundances of the 171 higher-quality metagenome assembled genomes (MAGs).** Abundance values have been scaled by the fraction of the DNA that mapped to the MAGs. The number of MAGs per lineage is provided in the row labels by 'n = #'.

## Identification of P450s in metagenomes

Across the six samples, 544,659 proteins were predicted from Armutlu, 3,097,365 from Balya, 5,495,569 from Gömeç, 1,042,240 from Hisaralan, 5,046,949 from Acıgöl and 1,881,283 from Tuz Gölü ([Fig 5](); total of 1,958,703 proteins after clustering at 100% identity) representing a substantial reservoir of potentially useful extremophilic biomolecules. An initial screening of the full protein dataset, conducted using a combination of HMM profile searches and alignment with reference sequences from the CYPED database, identified hundreds of putative cytochrome P450 enzymes. The distribution across the six sites was as follows: 55 from Armutlu, 614 from Balya, 801 from Gömeç, 434 from Hisaralan, 579 from Acıgöl, and 541 from Tuz Gölü. Before classifying these putative P450s, amino acid sequences were filtered to those with complete sequences (including both start and stop codons) and were searched against the CYPED database to eliminate non-microbial sequences. Those with at least a 20% match to microbial P450s were then examined for the integrity of their heme-binding domains using the NCBI CDD (Conserved Domain Database), and those that did not contain the consensus heme binding motif $F(x)_nG/A(x)_mCxG$ were removed (where: x is any amino acid; n is typically 2 but up to 5 in some families, e.g., CYP152; and m is typically 3 but up to 6 in some families [77]). After filtering, a total of 311 sequences remained: 52 thermophilic (Armutlu n = 3, Hisaralan n = 49), 92 acidophilic (Balya), and 167 halophilic (Gömeç n = 57, Lake Acıgöl n = 31, and Tuz Gölü = 79) ([Fig 5]()). Among these sequences, 241 were found across 138 of the bins (104 bins with a taxonomic classification at the class level or below), including 45 of the higher-quality MAGs ([S7 Table]()).

We did not observe a clear correlation between microbial diversity and either the number of P450s or the number of P450 families present in the samples ([S2 Fig]()). P450s from Balya, which had a relatively high microbial diversity (Shannon diversity of ~5.5), were only encoded by members of the phylum *Pseudomonadota*. Notably, ten of the higher-quality Balya MAGs encoded multiple P450s (from different P450 families; [S7 Table](), [S3 Fig]()). This included five members of the genus *Novosphingobium* that each encoded 4–9 P450s, and a *Blastomonas fulva* that encoded 7. At the hypersaline sites, Tuz Gölü and Gömeç (diversities of 3.7 and 6.5, respectively), members of the phylum *Halobacteriota*, specifically the families *Haloarculaceae*, *Halobacteriaceae*, *Haloferacaceae*, and *Salinarchaeaceae,* were the primary encoders with

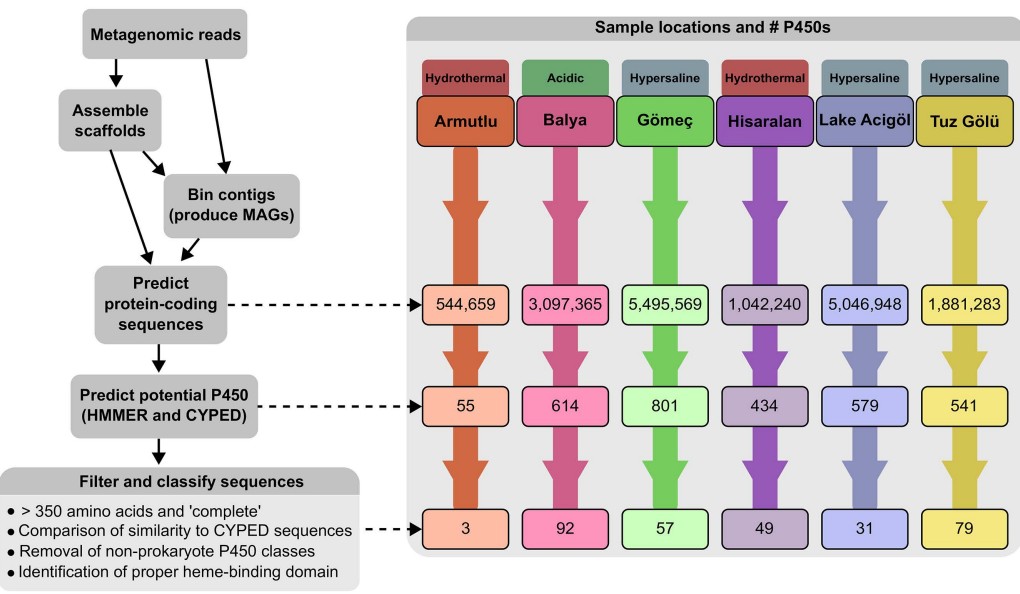

**Fig 5. Flowchart providing an overview of the sample processing steps, and the number of proteins and putative P450s obtained from each sample.**

1–5 P450s each. At the other hypersaline site, Lake Acıgöl (diversity of 5.8), P450s were primarily encoded by members of the phyla *Bacteroidota*, *Halobacteriota*, and *Pseudomonadota* (1–2 P450s). Among the hydrothermal sites Hisaralan and Armutlu (diversities of 4.0 and 5.4, respectively) bins were only obtained from Hisaralan, and the top encoders with 2–4 P450s included members of the phyla *Actinomycetota*, *Bacillota*, *Chloroflexota*, and *Desulfobacterota*_B.

## Classification of P450s

The 311 P450s were named according to P450 nomenclature criteria [71], with those having less than 40% amino acid identity designated as a new family, and those with more than 40% but less than 55% identity assigned to a new subfamily (Fig 6). The site with the highest number of identified P450s was Balya acid mine drainage (n = 92), followed by Tuz Gölü (79), Gömeç (57), Hisaralan (49), and Lake Acıgöl (31). Only three P450s were identified in the Armutlu hot water sample, possibly due to low DNA read depth obtained from sequencing, however all three belonged to different families, one

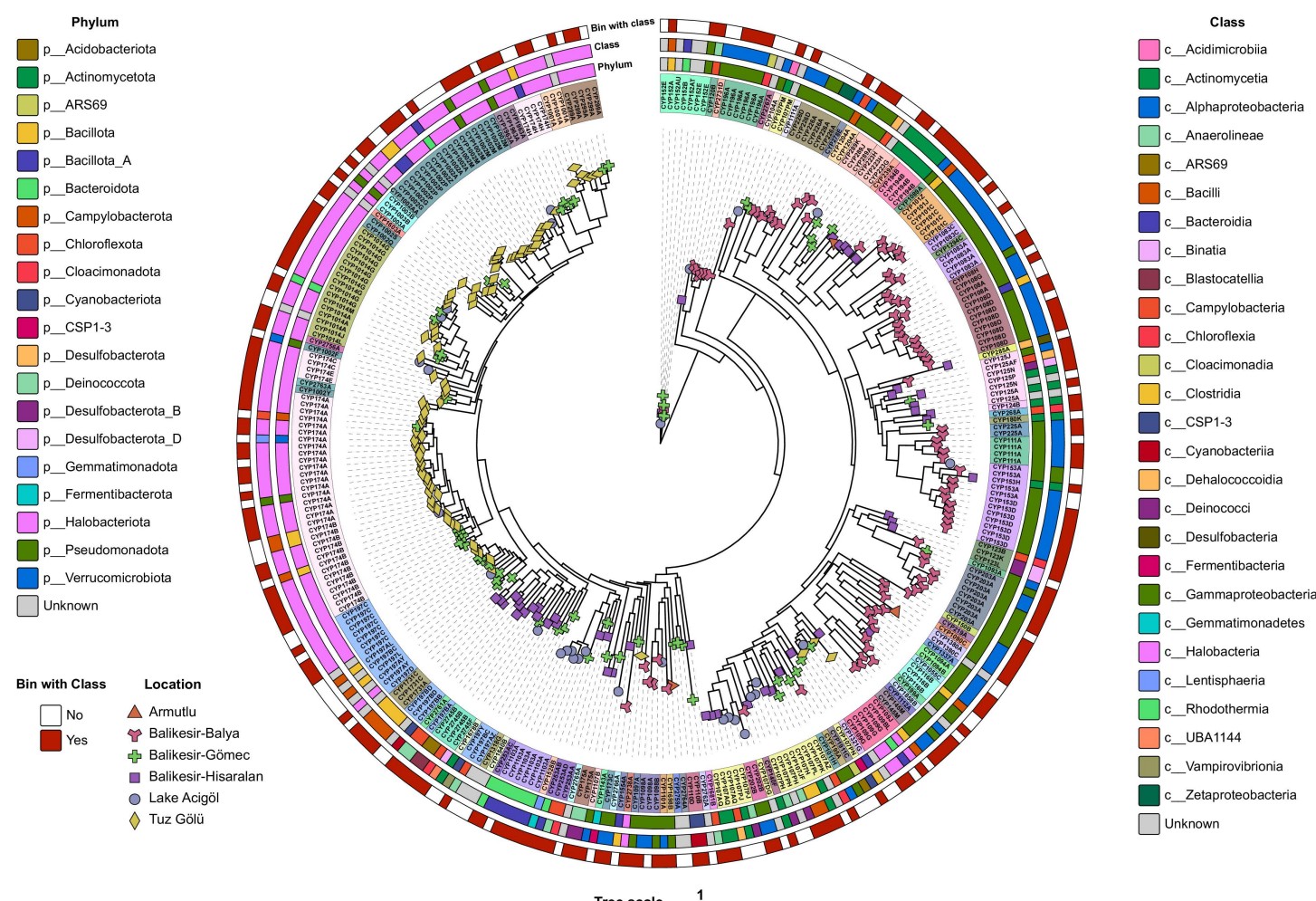

**Fig 6. Maximum likelihood tree of the 311 cytochrome P450 enzymes identified in the six samples.** The inner ring indicates the corresponding P450 subfamily (text) and family (highlight colour). The taxonomic classification of the host bin at the phylum and class level is shown in the second and third rings. For those proteins that were not found in one of the metagenomic bins or were in a bin with a poorly resolved taxonomic classification (not below the class level; see "Bin with class"), the closest P450 match in the GTDB reference genomes was used as a proxy where possible.

of which represented a novel subfamily. Aside from Armutlu, samples with the highest P450 family diversity were Balya (n = 37), Gömeç (27), Hisaralan (23), and Lake Acıgöl (16). While Tuz Gölü harbored the second highest number of P450s of the six samples, it had the lowest diversity (13) (Table 1).

The family, subfamily, and potential functional characteristics of the identified P450s for each site are presented in S8 Table. In total, 311 putative microbial cytochrome P450 enzymes were identified and classified into 87 families and 158 subfamilies including 8 new families from all sites except Acıgöl and 49 new subfamilies except Armutlu (Table 2). Notably, Gömeç and Hisaralan exhibited the highest number of newly identified families and subfamilies. Three self-sufficient P450s (CYP116B304 from Hisaralan, CYP116B171 and CYP116B21 from Balya) [78,79] and seven P450s from the CYP152 family typically associated with peroxygenase activity [80] were identified across Hisaralan, Gömeç, Acıgöl, and Balya. Notably, 54% of the P450 families identified in this study have not been previously identified.

The CYP107 family was present in all sample sites except Armutlu, while there were the following numbers of site-specific P450 families: 6 for Tuz Gölü, 7 for Acıgöl, 14 for Gömeç, 12 for Hisaralan, 28 for Balya, and 1 for Armutlu. Dominant P450 families across the samples were CYP174 in Tuz Gölü (n = 28) and Gömeç (n = 11), CYP1103 in Acıgöl (n = 6), CYP107 and CYP197 in Hisaralan (n = 7), and CYP108 in Balya (n = 12) (Fig 7).

The sequences of CYP107, the common family in all extreme sites, were found in nine of the bins with taxonomic classifications at the class level or below (three higher-quality MAGs), from the following phyla: *Actinomycetota* (genera JAHWLC01 and *Blastococcus* and order *Nitriliruptorales*); CSP1–3 (genus HRBIN32); *Deinococcota* (genus JAABTL01); *Bacillota* (genera YIM-78166 and *Ectobacillus*); and *Chloroflexota* (genus *Roseiflexus*). A total of nine, four of which are novel, were identified from hypersaline environments (CYP107PH1, CYP107PH2, CYP107PH3, CYP107PJ1,

**Table 1. Comparison of the main features of P450s among extreme sites.**

| | Armutlu | Balya | Gömeç | Hisaralan | Lake Acıgöl | Tuz Gölü |
|---|---|---|---|---|---|---|
| **Extreme condition** | Hydrothermal | Acidic | Hypersaline | Hydrothermal | Hypersaline | Hypersaline |
| **No. of P450s** | 3 | 92 | 57 | 49 | 31 | 79 |
| **No. of families** | 3 | 37 | 27 | 23 | 16 | 13 |
| **No. of subfamilies** | 3 | 46 | 45 | 50 | 18 | 29 |
| **Dominant P450 families** | | CYP108 | CYP174 | CYP107&CYP197 | CYP1103 | CYP174 |
| **P450 diversity percentage (%)*** | 100 | 40.2 | 47.4 | 46.9 | 51.6 | 16.5 |

* P450 diversity percentage was calculated as 100 x (Total number of P450 families/ Total number of P450s)

**Table 2. Classifications of P450s belonging to new families and subfamilies.**

| Site | Extreme Condition | Member of new Family | Member of new Subfamily |
|---|---|---|---|
| **Armutlu** | Hydrothermal | CYP2759A1 | – |
| **Balya** | Acidic | CYP2766A1, CYP2767A1 | CYP1055C1, CYP1294C1, CYP145M1, CYP1698B1, CYP1858B1, CYP278E1, CYP289K1 |
| **Gömeç** | Hypersaline | CYP2762A1, CYP2764A1, CYP2765A1 | CYP1002AA1, CYP1011H1, CYP107PK1, CYP107PL1, CYP107PM1, CYP1318G1, CYP1321G1, CYP1528B1, CYP152AT1, CYP152AU1, CYP1540B1, CYP180K1, CYP1911C1, CYP197AY1, CYP223H1, CYP253AC1, CYP253AD1, CYP2745F1 |
| **Hisaralan** | Hydrothermal | CYP2761A1 | CYP101Z1, CYP107PN1, CYP123K1, CYP123L1, CYP125AF1, CYP153H1, CYP1681B1, CYP1731C1, CYP197AZ1, CYP197BA1, CYP197BB1, CYP197BC1, CYP197BD1, CYP197Y1, CYP253AB1 |
| **Lake Acıgöl** | Hypersaline | – | CYP107PJ1, CYP1678B1, CYP197BC1, CYP2731D1, CYP289J1 |
| **Tuz Gölü** | Hypersaline | CYP2763A1 | CYP1002Y1, CYP1002Z1, CYP1014M1, CYP109BL1 |

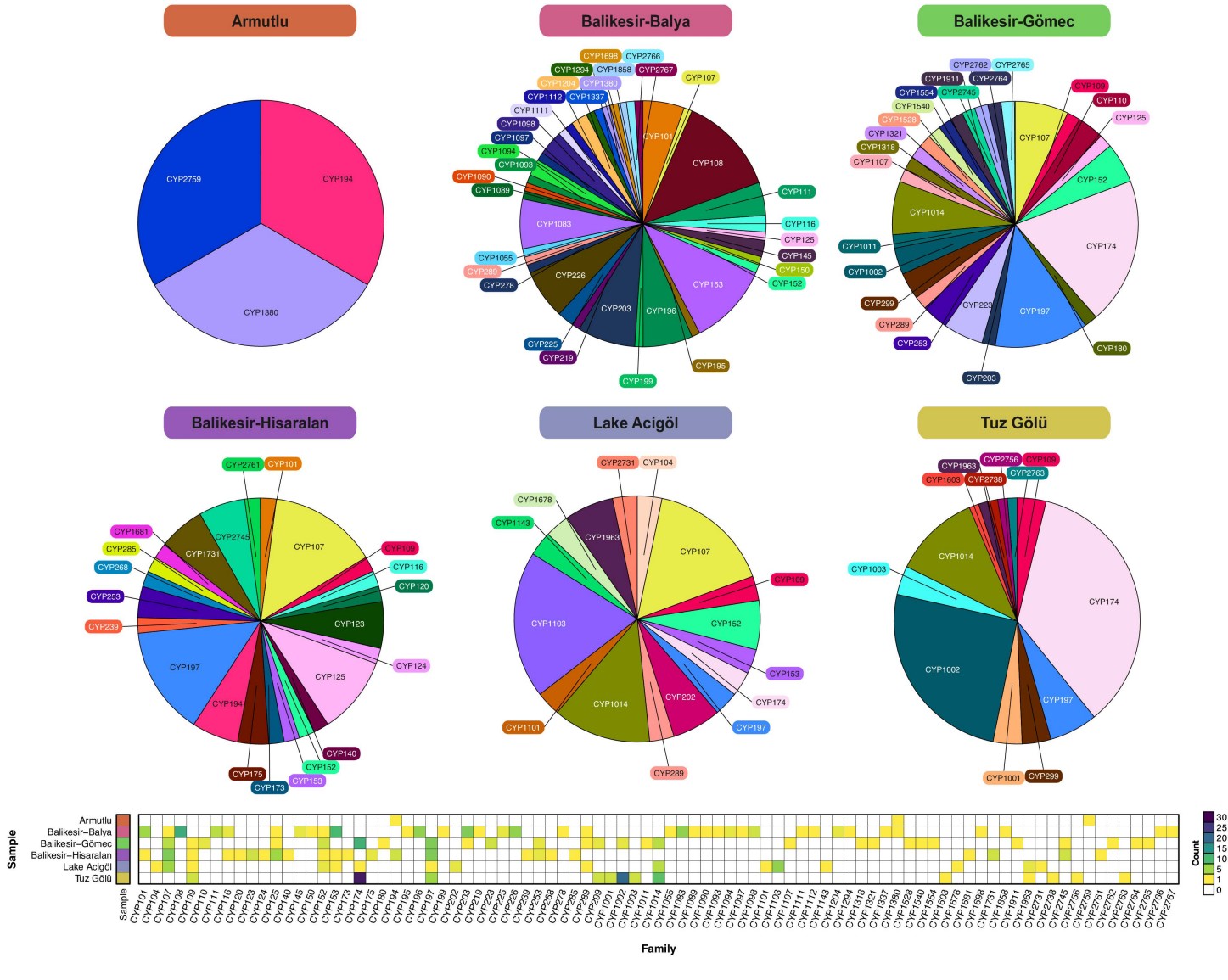

**Fig 7. Pie-graphs and heatmap showing the distribution and counts of the P450 families identified across the six sites.**

CYP107PJ2, CYP107PK1, CYP107PL1, CYP107PM1, and CYP107PM2), seven from the hydrothermal environments (CYP107AQ30, CYP107AQ31, CYP107AQ32, CYP107AZ2, CYP107H11, CYP107JF5, and CYP107PN1), and one from the acidic environment (CYP107DG12).

The CYP174 and CYP109 families were commonly found in the hypersaline habitats: CYP174B72 and CYP109G34 from Lake Acıgöl; CYP174A, CYP174B, CYP174C, CYP174E, CYP109BL1, CYP109G35 and CYP109G36 from Tuz Gölü; and CYP174A, CYP174B, CYP174E and CYP174H, CYP109G37 from Gömeç. All identified hosts of the CYP174 sequences belonged to the phylum *Halobacteriota* (28 bins, 10 higher quality MAGs), across the genera *Haloarchaeobius*, *Haloarcula*, *Halobacterium*, *Halobaculum*, *Halolamina*, *Halomicrobium*, *Halonotius*, *Haloplanus*, *Halorientalis*, *Halorubrum*, *Halosimplex*, *Natronomonas*, QS-5-70-15 (family *Haloarculaceae*), *Salinibaculum* and *Salinigranum* (S7 Table). The previously identified CYP174s, the CYP174A and CYP174B subfamilies have been ascribed to archaea [12], as found here.

The CYP109 sequences from hypersaline environments were found in archaeal bins from the phylum *Halobacteriota*, including the genera *Halorientalis* and *Salinarchaeum* (S7 Table).

In the hydrothermal site, Hisaralan, CYP197 (n = 7) and CYP125 (n = 5) were also common (Fig 7). Novel CYP197 subfamilies identified in Hisaralan included: CYP197AZ, CYP197BA, CYP197BB, CYP197BC, CYP197BD and CYP197Y. Across all samples, CYP197 sequences were present in eight bins with taxonomic classifications at the class level or below (including seven higher quality MAGs) from archaeal phyla (*Halobacteriota*: *Halolamina* and the *Salinigranum* genera, and the *Halobacteriaceae* family) and bacterial phyla (*Bacillota*, RAOX-1 (family); and *Chloroflexota*, CP2-2F and the JANWYT01 genus). CYP125 sequences were found across five taxonomically classified bins (two higher quality MAGs) from the phyla *Actinomycetota* (*Blastococcus* genus), *Chloroflexota* (HRBIN24 genus), *Desulfobacterota*_B (HRBIN30 genus), and *Pseudomonadota* (*Rhizorhabdus*, genus).

Balya acid mine drainage was one of the sites with the highest P450 diversity, with families CYP108 (subfamilies CYP108A, CYP108D, CYP108G and CYP108H) and CYP153 (subfamilies CYP153A and CYP153D) the most abundant in this area. Among the eight classified bins (six high quality MAGs) encoding CYP108 sequences, all were members of the phylum *Pseudomonadota* (genera *Erythrobacter*, *Sphingobium*, *Blastomonas*, *Novosphingobium*, and *Hydrogenophaga*). On the other hand, CYP153 sequences were found across seven bins (six higher quality MAGs) belonging to the phyla *Actinomycetota* (*Blastococcus* genus) and *Pseudomonadota* (*Blastomonas* and *Novosphingobium* genera).

## Discussion

Extreme environments can support diverse, extremophilic microbial communities that have developed unique adaptive strategies to survive, and that can encode enzymes with novel structural and functional properties. Among such enzymes are cytochromes P450, a highly diverse superfamily of enzymes capable of catalyzing a broad range of reactions, including aromatic and aliphatic hydroxylation, heteroatom oxidation, epoxidation, and dealkylation at N-, O- and S-centers [81]. The ability of P450s to transform structurally diverse compounds such as fatty acids, steroids, terpenes, and aromatic hydrocarbons makes them key players in microbial metabolism. The products of these reactions are of value to industry, including in the pharmaceutical, bioremediation, and fine chemical sectors [82].

Although no clear correlation was observed between the microbial diversity (Shannon index) and the number of P450s (and P450 families) across the samples, microbial composition did have a strong influence on both the number and diversity of P450 enzymes. Specifically, members of *Pseudomonadota* and *Actinomycetota* contributed a wide variety of P450 families, including CYP108, CYP109, and CYP153, and often encoded multiple P450s in their respective genomes. In the hypersaline environments, *Halobacteriota* species often encoded CYP174s, while *Bacillota* and *Chloroflexota* were linked to CYP125 and CYP197 diversity in hydrothermal habitats. These results suggest that specific microbial groups may shape the diversity of P450s in extreme environments more than overall microbial diversity.

CYP107 sequences were found in the samples from all extreme conditions. Among the most extensively studied CYP107s are those involved in antibiotic biosynthesis. CYP107A1 (P450eryF) from *Saccharopolyspora erythraea* contributes to erythromycin biosynthesis [83]. CYP107L1 from *Streptomyces venezuelae* is integral in the production of pikromycin, neomethymycin, novamethymycin, neopikromycin, and novapikromycin [84]. *Micromonospora griseorubida* CYP107E1 (MycG) is associated with mycinamicin biosynthesis [85], *Streptomyces himastatinicus* CYP107B (HmtT) with himastatin [86,87], and *Streptomyces thermotolerans* CYP107C1 (orfA) with carbomycin [88]. *Streptomyces avermitilis* CYP107W1 [89] and *Streptomyces sp.* 307-9 CYP107FH5 (CYP TamI) [90] are involved in oligomycin and triandamycin biosynthesis, respectively. Additionally, other CYP107 forms are involved in the biosynthesis of other natural products of use in medicine: CYP107Z14 from *Sebekia benihana* contributes to the synthesis of the immunosuppressant cyclosporin A [91]; *Streptomyces hygroscopicus* CYP107G1 plays a role in the biosynthesis of the antifungal and antitumor agent rapamycin [92]; and *Streptomyces sp.* SN-593 CYP107E6 is associated with the biosynthesis of reveromycin T [93], used in osteoporosis treatment. CYP107H1 (P450Biol) from *Bacillus subtilis* plays a pivotal role in the synthesis of pimelic

acid [94], a component involved in biotin synthesis, while CYP107BR1 (P450vdh) from *Pseudonocardia autotrophica* is engaged in vitamin D biosynthesis [95] and *Streptomyces avermitilis* CYP107X1 operates in the progesterone biosynthetic pathway [96]. CYP107 forms are also potentially useful for the detergent industry due to roles in glycocholic acid biosynthesis, as exemplified by *Streptomyces coelicolor* CYP107U1 [97]. While information on reactions, substrates, and products is available for the CYP107H subfamily (CYP107H1), the reactions catalyzed by the rest of the CYP107s identified in this study and their biotechnological significance are unknown. However, the predominance of CYP107 forms in the biosynthesis of complex secondary metabolites such as antibiotics suggests that the novel forms identified here may be useful in the search for new or modified antimicrobial agents or other natural products that may have useful properties.

The CYP174 and CYP109 families were widespread in hypersaline sites in this study. From a single previous study, CYP174 has been associated with terpene metabolism [98] but it is unclear whether this activity is common to other CYP174 family members. By contrast, there are many characterized CYP109s. A study of 128 *Bacillus* species identified the CYP109 family as the third most abundant P450 family [99], predicted to be involved in the synthesis of a wide range of secondary metabolites important to the physiology of *Bacillus* species. Among the characterized CYP109s, CYP109B1 from *Bacillus subtilis* strain 168 was found to be responsible for the hydroxylation of saturated fatty acids (C10-C18), methyl esters of saturated fatty acids (C12-C16), ethyl esters of saturated fatty acids (C12-C14) and unsaturated fatty acids (C14-C16). In addition to fatty acids, CYP109s can carry out the hydroxylation of primary n-alcohols (1-decanol, 1-dodecanol, and 1-tetradecanol) and the oxidation of the terpenes, α-ionone, β-ionone and (+)-valencene, which have an important place in the perfume, cosmetics, pharmaceutical, and other fine chemical industries [100,101]. Studies of CYP109E1 from *Bacillus megaterium* DSM319 have demonstrated that this enzyme can hydroxylate testosterone and vitamin D3 to synthesize industrially valuable products [102,103]. In addition, the CYP109E1 enzyme is capable of the hydroxylation of statins (compactin, lovastatin, and simvastatin) to synthesize drug metabolites and the hydroxylation of terpenes (α-ionone, β-ionone, nootkatone, isolongifolen-9-one, α-damascone, β-damascone, and β-damascenone) to synthesize valuable terpene derivatives with high regioselectivity [104]. Together with CYP109E1, CYP109A2—another CYP109 from *B. megaterium* DSM319—was found to hydroxylate vitamin D3 with high regioselectivity [105]. In addition to CYP109s from *Bacillus* species, studies with *Sorangium cellulosum So ce56* showed that the organism has three CYP109s: CYP109C1, CYP109C2, and CYP109D1. CYP109D1 and CYP109C2 are responsible for the hydroxylation of lauric acid (C12), tridecanoic acid (C13), myristic acid (C14), and palmitic acid (C16), whereas CYP109D1 can also hydroxylate capric acid (C10) [106,107]. These studies suggest that the CYP109 family can catalyze many different reactions and substrates. Notably, the CYP109 and CYP174 members identified in this study are from subfamilies that have not been characterized in any detail biochemically. While the known substrate profiles of these families do not directly indicate roles in salt adaptation, their prevalence across the hypersaline microbiomes suggests they may possess structural features enabling function under high-salinity conditions. These observations highlight the need for future functional and structural studies to explore their potential halotolerance and biotechnological relevance.

The CYP125 and CYP197 families were also common in the hydrothermal site, Hisaralan. Previously characterized members of the CYP125 family are CYP125A6 and CYP125A7, which play a role in steroid hydroxylation pathways and in cholesterol catabolism in mycobacterial species [108]. These enzymes may also be linked to membrane lipid composition and ordering in thermophiles, which can be influenced by cholesterol across a wide temperature range [109,110]. Based on this information, it can be hypothesized that members of the CYP125 family, including the newly identified CYP125N, CYP125P, and CYP125AF subfamilies, may contribute to pathways that facilitate microbial adaptation to high temperatures in hydrothermal environments. Members of the CYP197 family have been found across various bacterial phyla and are frequently encoded within biosynthetic gene clusters associated with secondary metabolism [11,99]. While their specific enzymatic functions and underlying catalytic mechanisms remain uncharacterized, their presence in both hydrothermal sites suggests a role in the biosynthesis of heat-stable or stress-responsive metabolites. Functional characterization of these enzymes may uncover novel biocatalysts with potential applications in biotechnology and natural product discovery.

CYP108 was one of the two dominant P450 families at the acid mine drainage site, Balya. While there is limited research on CYP108, members of this family are known to catalyze the oxidation of α-terpineol [111]. For example, the CYP108D1 enzyme exhibits hydroxylase activity on aromatic hydrocarbons, including phenyl cyclohexane and p-cymene [112]. CYP153 was also common to the Balya site; this family has been associated with alkane degradation in diverse bacterial species, including members of the phyla *Actinobacteria* (now *Actinomycetota*) and *Proteobacteria* (now *Pseudomonadota*) [113–116]. To date, the best characterized members of the CYP153 family include CYP153A6 from *Mycobacterium* sp. HXN-1500, which hydroxylates medium-chain-length alkanes (C6 to C11) to 1-alkanols [117], and CYP153A13 from *Alkanivorax borkumensis* SK2. CYP153A13 has diverse catalytic capability, being able to hydroxylate not just the terminal end of short alkyl groups attached to aromatic rings but also the *p*-position of phenolic compounds substituted with a halogen or an acetyl group. Additionally, CYP153A13a demonstrated the ability to demethylate aromatic compounds containing methyl ether groups [118]. Organic compounds, including aromatic hydrocarbons and alkanes, are present in acid-mine drainage sites like Balya. Therefore, microorganisms from such sites, and the enzymes encoded in their genomes, may be useful for degrading these hydrocarbons [119]. Considering all the information known about the CYP108 and CYP153 families, undertaking further in-depth studies on the P450s from the Balya site to elucidate the hydrocarbon groups they degrade holds promise for advancing bioremediation initiatives.

Finally, rare exceptions to the $F(x)_nG(x)_mCxG$ motif used for filtering sequences have been described previously [77], where one or more of the specified residues is conservatively substituted. However, the Cys residue is almost universally conserved and generally considered to be required for generating the highly reactive oxidizing species, compound I, involved in monooxygenase activity. Notably, among the sequences excluded based on the heme-binding motif was a CYP102A178 sequence that appeared to encode a plausible P450 sequence, with the exception that the conserved Cys was replaced by a Tyr residue. Further work is underway to characterise both the putative Tyr and Cys forms of this enzyme.

## Conclusion

Metagenomics is a powerful tool for discovering novel biocatalysts from uncultured microorganisms. Through shotgun metagenomics and computational analyses, this study has identified 311 P450 sequences, including 8 novel families and 49 subfamilies, from diverse extreme environments across Türkiye. Of these sequences, 237 were associated with 138 metagenomic bins or metagenome assembled genomes (MAGs) of prokaryotic extremophiles, many representing taxonomically novel lineages. These findings underscore the untapped microbial diversity in Türkiye's extreme environments and their potential as rich reservoirs for novel biocatalysts with applications in industrial and environmental biotechnology.

The taxonomic and P450 diversity uncovered in this study contributes to the growing catalogue of reference data for extremophilic microorganisms and their enzymes. These data can support the development of environment-specific microbial or enzymatic markers, aiding the identification of samples from similar geochemical conditions. Previous studies have shown that metagenomic data carry distinctive environmental signatures; for example, they have been used to infer the geographic origin of ancient samples [120], map the spatial distribution of antimicrobial resistance [121], and classify environments using machine learning models [122,123]. By contributing new reference data and uncovering novel P450 lineages, this study provides a valuable resource for future research into the ecological roles and biotechnological potential of extremophile-derived enzymes.

## Supporting information

**S1 Fig. Nonpareil curves.**
(TIF)

**S2 Fig. Shannon diversity versus number of P450s and P450 families.**
(TIF)

**S3 Fig. Number of P450s and P450 families per bin.**
(TIF)

**S1 Table. Bin abundances.**
(XLSX)

**S2 Table. Nonpareil results.**
(XLSX)

**S3 Table. SingleM abundances.**
(XLSX)

**S4 Table. Read stats.**
(XLSX)

**S5 Table. All bins summary.**
(XLSX)

**S6 Table. Number of phyla per sample.**
(XLSX)

**S7 Table. Sample P450 data.**
(XLSX)

**S8 Table. P450s that were obtained from the six extremophile sample sites and the functions, where known, of previously characterized members of the same P450 family.**
(DOCX)

## Author contributions

**Conceptualization:** Hande Mumcu, Nevin Gul Karaguler.

**Data curation:** Julian Zaugg, Philip Hugenholtz.

**Formal analysis:** Julian Zaugg, David R. Nelson, Philip Hugenholtz.

**Funding acquisition:** Nevin Gul Karaguler.

**Investigation:** Hande Mumcu, Irem Keles, Aycan Kayrav, Nevin Gul Karaguler.

**Methodology:** Hande Mumcu, Julian Zaugg, Irem Keles, Aycan Kayrav, Philip Hugenholtz, Elizabeth M. J. Gillam, Nevin Gul Karaguler.

**Project administration:** Nevin Gul Karaguler.

**Resources:** Julian Zaugg, Nurgul Balci, David R. Nelson, Philip Hugenholtz.

**Software:** Julian Zaugg, David R. Nelson, Philip Hugenholtz.

**Supervision:** Nevin Gul Karaguler.

**Validation:** Hande Mumcu, Julian Zaugg, Irem Keles, Aycan Kayrav, David R. Nelson, Philip Hugenholtz, Elizabeth M. J. Gillam, Nevin Gul Karaguler.

**Visualization:** Hande Mumcu, Julian Zaugg, Nevin Gul Karaguler.

**Writing – original draft:** Hande Mumcu, Julian Zaugg, Irem Keles, Aycan Kayrav, Nevin Gul Karaguler.

**Writing – review & editing:** Hande Mumcu, Julian Zaugg, Irem Keles, Aycan Kayrav, Nurgul Balci, David R. Nelson, Philip Hugenholtz, Elizabeth M. J. Gillam, Nevin Gul Karaguler.

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
