## [Decision Letter · Decision Letter 0]

21 Jan 2025

PONE-D-24-34417Exploring extreme environments in Türkiye for novel P450s through metagenomic analysisPLOS ONE

Dear Dr. Gül Karagüler,

Thank you for submitting your manuscript to PLOS ONE. After careful consideration, we feel that it has merit but does not fully meet PLOS ONE’s publication criteria as it currently stands. Therefore, we invite you to submit a revised version of the manuscript that addresses the points raised during the review process.

The following revisions are compulsory:

1. It must be specified in the methodology section under a separate section called data records where the data has been deposited such as the NCBI with the required ascension numbers.

2. It is unclear if any custom designed codes have been used.  In the methods section there must be a section for code availability in which the authors must specify all the custom designed codes that was used.  If none was used, this must be specified.

3. All figures must be rechecked to comply with the PLOS one illustration standards for publication.

4. There must be an overview of methods implemented at the beginning of the methods section in the form of a schematic representation.

5.  The entire manuscript must be checked for English and grammar.  It is recommended that this is done by a professional language editor.

6.  All comments by all reviewers must be addressed in a separate word document in the following format:

Reviewer 1 (the reviewer number must be specified)

Reviewer comment (as specified by the reviewer)

Authors response (your response and justification to the reviewer's comment)

Changes to manuscript (Specific changes to manuscript if any.  If none is made it must be specified) Note that revision number 6 specified above must be done in this manner to assess if the manuscript has met the PLOS publication standards.  Non-compliance will lead to delays in assessing your manuscript.==============================

We look forward to receiving your revised manuscript.

Kind regards,

Preenan Pillay

Academic Editor

PLOS ONE

2. Thank you for stating the following financial disclosure:  [1- NGK, Grant No. 1059B192100859, The Scientific and Technological Research Council of Turkey (TUBITAK), https://tubitak.gov.tr/en

2- NGK, Grant Nos. 42997 and 42953, ITU Scientific Research Projects Division, https://bap.itu.edu.tr/]. 

3. Please note that your Data Availability Statement is currently missing [the repository name and/or the DOI/accession number of each dataset OR a direct link to access each database]. If your manuscript is accepted for publication, you will be asked to provide these details on a very short timeline. We therefore suggest that you provide this information now, though we will not hold up the peer review process if you are unable.

Additional Editor Comments:

Thank you for submitting your article to PLOS one. The article presents findings which appear to be novel however there are several critical areas that need to be addressed to be accepted for publication. They are as follows:

1. It must be specified in the methodology section under a separate section called data records where the data has been deposited such as the NCBI with the required ascension numbers.

2. It is unclear if any custom designed codes have been used. In the methods section there must be a section for code availability in which the authors must specify all the custom designed codes that was used. If none was used, this must be specified.

3. All figures must be rechecked to comply with the PLOS one illustration standards for publication.

4. There must be an overview of methods implemented at the beginning of the methods section in the form of a schematic representation.

5. The entire manuscript must be checked for English and grammar. It is recommended that this is done by a professional language editor.

6. All comments by all reviewers must be addressed in a separate word document in the following format:

Reviewer 1 (the reviewer number must be specified)

Reviewer comment (as specified by the reviewer)

Authors response (your response and justification)

Changes to manuscript (Specific changes to manuscript if any)

Reviewers' comments:

Reviewer's Responses to Questions

**Comments to the Author**

1. Is the manuscript technically sound, and do the data support the conclusions?

Reviewer #1: Yes

Reviewer #2: Yes

Reviewer #3: Yes

2. Has the statistical analysis been performed appropriately and rigorously? 

Reviewer #1: I Don't Know

Reviewer #2: Yes

Reviewer #3: Yes

3. Have the authors made all data underlying the findings in their manuscript fully available?

Reviewer #1: Yes

Reviewer #2: Yes

Reviewer #3: Yes

4. Is the manuscript presented in an intelligible fashion and written in standard English?

Reviewer #1: Yes

Reviewer #2: No

Reviewer #3: Yes

5. Review Comments to the Author

Reviewer #1: 1. The figure images are blurry. Improve.

2. How many samples were collected?

3. Check the MS for minor typos. See line 219, there should be a full stop before the word "Notably." Also, check the sentence in line 261, its confusing.

4. State the extreme conditions for the sites in Table 1, just like in Table 2.

5. Table 2: Provide references for previously characterized members. Table 2 is too long and should be taken to Supplementary Section.

6. There is no caption for the conclusion section. If this is the journal's style, then ignore.

7. The conclusion should be concise and limited to the scope of the study. The conclusion should properly align with the stated objective of the work.

8. The authors may revise the introductory section to highlight the significance of metagenomics for environmental forensics in general terms, thereby giving a general background. The current version seem too focused. See the following: https://doi.org/10.1016/j.mib.2015.05.005, https://doi.org/10.1007/s11157-019-09501-4, https://doi.org/10.1007/978-3-319-97922-9_4, https://doi.org/10.1038/nrg1709, https://doi.org/10.3389/fenvc.2023.1052697, etc.

Reviewer #2: The authors present findings on the characterization of microbial communities and the associated P450 enzyme diversity from six extreme environmental sites across Türkiye, employing de novo sequence binning, phylogenetic analysis, and functional gene annotation of metagenomic data. Their work led to the discovery of eight new P450 families and 49 new subfamilies.

While the study aligns well thematically and provides a valuable contribution, certain sections lack sufficient depth and require revision before it can be considered for publication.

1. For the abstract,

- provide a more explicit statement of the research objective and its broader significance to industrial applications.

- It would be helpful to clarify whether the study confirmed any specific enzymatic activities or if the focus was solely on diversity and classification.

- consider rephrasing the final sentence to reinforce the practical implications of the study, linking enzyme diversity to specific future applications.

2. The information in the introduction could be better organized to create a more cohesive narrative that seamlessly leads to the study’s aim. See specific comments below,

- The study's aim is introduced only at the end of the introduction. Consider weaving this aim into the earlier discussion to provide context and create a clear research trajectory.

- Line 50-61: Not sure whether/how these information is relevant to this specific study. The general discussion of P450 enzymes should be structured to create a logical progression from their fundamental properties and significance to the specific objectives of the study.

- Line 79: Provide examples for the numerous biotechnological applications mentioned here.

- Provide the limitations of previous studies and how this research addresses gaps in knowledge, particularly in relation to extremophilic P450s.

- The mention of metagenomics and -omics sciences is relevant, but the description could benefit from more detail about why these methods are particularly effective for uncovering novel P450 enzymes.

- The examples of characterized P450s (e.g., CYP152, CYP203) are helpful, but the connection to the present study could be more explicitly stated. For instance, how do these findings inspire or relate to the current research?

3. The labels in Figures 1–6, especially in Figures 2, 3, 5, and 6, are unclear and difficult, or impossible, to read. The formatting of all figures needs to be revised to ensure that the labels and data points are clearly visible and legible.

4. Lines 211-227: The significance of identifying dominant phyla and their environmental roles is mentioned briefly, but the implications for the study's objectives need to be elaborated. For example, discuss how the microbial community composition influences the diversity of P450 enzymes.

5. Provide a short summary/conclusion statement for first two the results sections: “Taxonomic profiling of the extreme sites, Metagenome assembled genomes (MAGs)”.

6. Revise the sentence in lines 302-303 “Notably, for 64 % of P450 families identified here no family member has yet been characterized functionally” for clarity.

7. Revise the title of Table 2 for improved clarity: "Table 2. Classified P450s and the functions were known of previously characterized members of the same family obtained from the six extremophile sample sites”.

8. Where possible, references and the source organisms for the known functions that are provided in Table 2.

9. Ensure that figure captions are not embedded within the main text. Instead, place each caption directly below its corresponding figure for better organization and readability.

10. Lines 357-359: The sentence is overly general and lacks specific details. Restructure the sentence providing specific examples, context, and evidence for "various industrial sectors" and "useful activities".

11. In lines 369–370, the sentence “Additionally, other CYP107 forms are involved in the biosynthesis of other natural products of use in medicine” requires elaboration. Provide examples of these compounds, their roles, and the medical conditions they are used to treat.

12. Lines 413–415 state: "Thus, the characterization of the CYP109 families and the elucidation of the functions of the subclasses of the CYP174 family defined in this study is of interest for future studies." Based on previously published data, speculate on the potential functions and advantages of the CYP109 families and CYP174 family subclasses defined in this study, particularly in the context of extremophilic environments.

13. Lines 423–424 state: “However, uncertainties remain regarding the other identified subfamilies in current study, namely CYP125N, CYP125P, and the newly discovered CYP125AF.” Clarify the intended meaning of this sentence. Does it refer to gaps in knowledge about the specific functions of these subfamilies, their potential roles in extremophilic environments, or both? Additionally, consider elaborating on the significance of addressing these uncertainties and how future studies might resolve them.

14. Lines 424–427 state: “On the other hand, the presence of CYP197 in different bacterial phyla has been associated with secondary metabolism [6, 83]; however, biochemical and functional characterization of this family is lacking.” Clarify the intended meaning of this sentence. Does it imply that while CYP197 has been linked to secondary metabolism, its specific roles or mechanisms remain unknown? Additionally, expand on why uncovering the functions of CYP197 in hydrothermal environments is significant. For example, consider how such discoveries could contribute to understanding extremophile adaptations, potential biotechnological applications, or the production of novel metabolites.

Reviewer #3: Minor Comments on the Article:

1. Presentation of Program Details:

In the text, when describing the programs used, both the version and the GitHub link are included in parentheses in several places (e.g., lines 142, 143, 147, 148, 154). Although this provides valuable information, the text would be clearer and easier to read if these details were moved to the bibliography or footnotes.

2. Formatting of Table 2:

Table 2 spans 12 pages, which makes it quite lengthy. It would be more user-friendly to compress the table. For example, the information in the "P450 Name" column does not necessarily need to be in separate rows. You could merge some rows and list the names separated by commas, provided the other data in those rows is consistent.

3. Description of Taxonomic Classification:

The authors classify metagenomic reads into various taxonomic groups. It would be beneficial to include a broader description of this classification approach. For example, references to relevant literature could provide additional context:

o Wajid, B., et al. "Music of metagenomics—a review of its applications, analysis pipeline, and associated tools [Erratum: February 2022, v. 22 (1); p. 137]." (2022).Taxometer: Improving taxonomic classification of metagenomics contigs

o Kawulok, J., and Deorowicz, D. "CoMeta: classification of metagenomes using k-mers." PloS one 10.4 (2015): e0121453.

o Ounit, R., et al. "CLARK: fast and accurate classification of metagenomic and genomic sequences using discriminative k-mers." BMC genomics 16 (2015): 1-13.

o Breitwieser, F. P., et al. "KrakenUniq: confident and fast metagenomics classification using unique k-mer counts." Genome biology 19 (2018): 1-10.

4. Applications of Environmental Metagenomics:

The authors highlight that various extreme environments can harbor organisms that produce new, unique P450 enzymes. It might be worth mentioning that this information could also be used in reverse—for example, to classify an unknown sample into a specific environment. Relevant studies include:

o Bozzi, Davide, et al. "Towards predicting the geographical origin of ancient samples with metagenomic data." Scientific Reports 14.1 (2024): 21794.

o Zhelyazkova, Maya, et al. "Origin sample prediction and spatial modeling of antimicrobial resistance in metagenomic sequencing data." Frontiers in Genetics 12 (2021): 642991.

o Kawulok, J. et al. "Environmental metagenome classification for constructing a microbiome fingerprint." Biology Direct 14 (2019): 1-23.

o Anyaso-Samuel, et al. "Metagenomic geolocation prediction using an adaptive ensemble classifier." Frontiers in Genetics 12 (2021): 642282.

6. PLOS authors have the option to publish the peer review history of their article (what does this mean? ). If published, this will include your full peer review and any attached files.

**Do you want your identity to be public for this peer review?** For information about this choice, including consent withdrawal, please see our Privacy Policy .

Reviewer #1: No

Reviewer #2: No

Reviewer #3: No

---

## [Author Response · Author response to Decision Letter 1]

29 Jul 2025

We would like to thank the editor and reviewers for their constructive feedback and suggestions, which helped us improve the quality and clarity of our manuscript. We have carefully addressed all comments and detailed our responses in the attached “Response to Reviewers” document. Additionally, the manuscript and supplementary materials have been revised accordingly.

We hope the revised version meets your expectations.

---

## [Editor Report · Decision Letter 1]

4 Aug 2025

Exploring extreme environments in Türkiye for novel P450s through metagenomic analysis

PONE-D-24-34417R1

Dear Dr. Gül Karagüler,

We’re pleased to inform you that your manuscript has been judged scientifically suitable for publication and will be formally accepted for publication once it meets all outstanding technical requirements.

Kind regards,

Preenan Pillay

Academic Editor

PLOS ONE

Additional Editor Comments (optional):

I would like the authors for taking the time to review the manuscript and improving the quality and data integrity of the manuscript. Based on the revisions done the manuscript is accepted for publication however during the publication process the authors must scan the manuscript for minor grammatical and scientific phrasing errors.
---

## [Editor Report · Acceptance letter]

PONE-D-24-34417R1

PLOS ONE

Dear Dr. Gül Karagüler,

I'm pleased to inform you that your manuscript has been deemed suitable for publication in PLOS ONE. Congratulations! Your manuscript is now being handed over to our production team.

Kind regards,

on behalf of

Prof Preenan Pillay

Academic Editor

PLOS ONE